# Simulated Galactic Cosmic Rays Modify Mitochondrial Metabolism in Osteoclasts, Increase Osteoclastogenesis and Cause Trabecular Bone Loss in Mice

**DOI:** 10.3390/ijms222111711

**Published:** 2021-10-28

**Authors:** Ha-Neui Kim, Kimberly K. Richardson, Kimberly J. Krager, Wen Ling, Pilar Simmons, Antino R. Allen, Nukhet Aykin-Burns

**Affiliations:** 1Center for Musculoskeletal Disease Research and Center for Osteoporosis and Metabolic Bone Diseases, Department of Internal Medicine, Division of Endocrinology and Metabolism, University of Arkansas for Medical Sciences, 4301 W. Markham Street, Little Rock, AR 72205, USA; KKRichardson@uams.edu (K.K.R.); WLing@uams.edu (W.L.); 2Department of Pharmaceutical Sciences, Division of Radiation Health, University of Arkansas for Medical Sciences, 4301 W. Markham Street, Little Rock, AR 72205, USA; KJKrager@uams.edu (K.J.K.); PGSimmos@uams.edu (P.S.); ARAllen@uams.edu (A.R.A.)

**Keywords:** galactic cosmic rays, space radiation, bone loss, osteoclast, osteoblast, mitochondria, redox, reactive oxygen species

## Abstract

Space is a high-stress environment. One major risk factor for the astronauts when they leave the Earth’s magnetic field is exposure to ionizing radiation from galactic cosmic rays (GCR). Several adverse changes occur in mammalian anatomy and physiology in space, including bone loss. In this study, we assessed the effects of simplified GCR exposure on skeletal health in vivo. Three months following exposure to 0.5 Gy total body simulated GCR, blood, bone marrow and tissue were collected from 9 months old male mice. The key findings from our cell and tissue analysis are (1) GCR induced femoral trabecular bone loss in adult mice but had no effect on spinal trabecular bone. (2) GCR increased circulating osteoclast differentiation markers and osteoclast formation but did not alter new bone formation or osteoblast differentiation. (3) Steady-state levels of mitochondrial reactive oxygen species, mitochondrial and non-mitochondrial respiration were increased without any changes in mitochondrial mass in pre-osteoclasts after GCR exposure. (4) Alterations in substrate utilization following GCR exposure in pre-osteoclasts suggested a metabolic rewiring of mitochondria. Taken together, targeting radiation-mediated mitochondrial metabolic reprogramming of osteoclasts could be speculated as a viable therapeutic strategy for space travel induced bone loss.

## 1. Introduction

The effects of high linear energy transfer (LET) radiation on human physiology and metabolism have been receiving more attention as NASA speeds up its deep-space human exploration plans. Exposure to galactic cosmic rays (GCR) and microgravity are two major hazards astronauts will face during long-duration spaceflights [1,2,3]. A reduction in bone mineral density is one of the most substantial deficits known to occur in astronauts during both short- and long-duration space travel and habitation [4,5,6,7]. Thus, it is necessary to understand the molecular mechanisms by which space travel induces bone loss to develop effective countermeasures for resolving this significant health risk.

Bone is a highly dynamic tissue, which is continuously remodeled by osteoclastic resorption and osteoblastic formation that occur in response to numerous biochemical and mechanical signals [8,9]. This reciprocal regulation of bone remodeling cells can be disrupted due to exposure to high LET radiation as well as exposure to microgravity during space travel. Numerous flight missions, as well as in vitro and in vivo studies, have documented microgravity as a challenge to bone, and more recently, high linear energy transfer (LET) radiation has been identified as causing bone damage [4,10,11,12,13,14,15,16].

All cell types that have been associated with bone remodeling are affected by radiation-induced cellular injury. The majority of the studies demonstrating the ionizing radiation effects on the bone and bone cells have been utilizing low LET radiation sources [17,18,19,20,21,22,23,24]. Although most of these studies focus on bone forming cells and point out that bone damage is mainly due to IR-induced defects in osteoblasts (OBs), the direct and indirect IR effects on osteoclasts (OCs) also were reported [19,25,26,27,28,29,30,31,32,33,34]. Increased OC numbers following IR exposures have been described by several groups implying that the main cell type responsible for radiation-induced bone loss are the OCs [11,22,28,35,36,37,38]. Briefly, these in vitro and in vivo studies demonstrated that X-rays or γ rays resulted in upregulation of OC marker genes, increased OC numbers and surface area as well as amplified osteoclastic activity that is apparent by the increased serum markers for bone resorption [11,22,35,36,37,38,39].

While the high and low LET radiation have different properties, the studies conducted using low doses of high LET radiation have similar findings, suggesting space travel-relevant radiation doses can induce bone loss [22,40,41,42,43,44]. These high LET studies have mostly focused on the bone resorbing OCs, proposing that increased osteoclastogenesis and increased osteoclastic activity are the major culprits for bone loss following radiation exposure. There is very little published literature on bone forming OBs in this regard, which reported that exposure to iron particles at low doses resulted in impaired OB development [45]. Despite their admirable well-thought experimental design and execution, comparing the results of these in vitro as well as in vivo studies published with high LET exposures has been difficult for assessing the effects of space radiation on bone tissue. The major limitation for reaching a consensus based on their findings is the use of different types of heavy ion particles (single ion or multiple ions sequentially delivered) in addition to studies using combined microgravity and radiation injury model [12,22,42,43,44,45,46,47,48,49,50,51,52,53]. Thus, it is desirable to have a consistent regimen of high LET exposure studies that would simulate the galactic cosmic rays (GCR) to tease out the molecular mechanisms affecting space radiation induced bone loss in the absence of microgravity.

IR can damage cells directly (i.e., single or double strand breaks on DNA) or indirectly through the generation of reactive oxygen species (ROS) [35,54,55]. Continuous exposure to low levels of high LET radiation could induce inflammatory processes due to increased secretion of pro-inflammatory cytokines, such as IL-1, IL-6 or TNF-α [11]. Such persistent inflammation, as well as IR-induced ROS, could stimulate the receptor activator of nuclear factor kappa-B ligand (RANKL) expression, which is an essential player for the development and activation of OCs [11,56]. Several molecular players in this process have been proposed in the field, including suppression of the transcriptional activity of FoxOs and increased intracellular H_2_O_2_, autophagy, ER stress and alterations in mitochondrial dynamics promoting OC differentiation [57,58,59,60].

In the current study, we primarily evaluated the effects of low dose simulated GCR exposure on bone tissue in adult male mice. Our results indicated that exposure to high LET space radiation increased trabecular bone loss via enhanced osteoclastogenesis. Furthermore, exposure to GCR elevated ROS levels and stimulated mitochondrial respiration in OCs differentiated from bone marrow precursor cells. These alterations were also accompanied by changes in metabolic pathways associated with mitochondrial function.

## 2. Results

### 2.1. GCR Induced Trabecular Bone Loss in Adult Mice

Despite widespread evidence that ionizing radiation leads to both an increase in osteoclasts and a decrease in osteoblasts and eventually causes bone diseases, the molecular mechanisms responsible for the adverse effects of high LET space radiation are not yet understood. To further investigate whether GCR may impair skeletal health and activate osteoclasts through mitochondrial mechanisms, we exposed 6-month-old male Balb/C mice to 0.5 Gy total body simplified GCR irradiation. Blood, cells and tissues were collected 3 months following IR exposure. As determined by micro-CT, GCR caused a dramatic decrease in trabecular but not cortical bone mass at the femur compared to sham controls (Figure 1A–C). This change was associated with a decrease in trabecular number and an increase in trabecular spacing while no changes were detected in trabecular thickness (Figure 1B).

In contrast to the femur, GCR had no effect on the trabecular bone mass of the spine in adult mice (Figure 2A,B). Taken together, these results indicate that GCR causes severe bone loss, and its effect is greater in the femur than the spine and more pronounced in trabecular than cortical bone. Consistent with this, we have shown earlier that distinct mechanisms are responsible for skeletal aging in cortical versus trabecular bone [61].

### 2.2. GCR Increased Bone Resorption Markers In Vivo and Osteoclast Maturation In Vitro

To gain insight into the cellular mechanism(s) of the adverse effect of GCR on skeletal health, we first performed histologic analysis to evaluate osteoclast and osteoblast numbers at the trabecular bone surface of the femur. Nine-month-old male mice showed no differences in the number or surface of both osteoclasts (Figure 3A,B) and osteoblasts (data not shown). However, the serum levels of C-terminal telopeptide of type 1 collagen (CTx)—a marker of bone resorption—in the GCR group trended higher than in sham controls (Figure 3C). Consistent with these changes, the expression of late/terminal osteoclast differentiation markers was increased in femur shafts of the GCR exposed mice (Figure 3D).

To probe into the mechanism of the increased resorption in the irradiated mice in the face of unchanged osteoclast numbers, we examined early or late differentiation of osteoclasts. Cultures of bone marrow macrophages from the GCR group formed slightly more osteoclasts than those from control mice (Figure 4A). The expression of late/terminal osteoclast differentiation markers was dramatically higher in irradiated mice than in control mice (Figure 4B). The mRNA levels of the factor known to induce early osteoclast differentiation, *Nfatc1* [62], were not affected by GCR (Figure 4C) 3 months after GCR exposure. In addition, GCR did not affect the mRNA levels of *Pgc1β*—a transcription factor that regulates mitochondrial biogenesis at this time point [63] (Figure 4C). Importantly, cultured osteoclasts from mice exposed to radiation had significantly higher mRNA levels of mitochondrial deacetylase *Sirt3*—an essential protein for osteoclast mitochondrial activity and bone resorption in the development of osteoporosis [64,65,66]—as compared to sham-irradiated mice (Figure 4C).

### 2.3. GCR Had No Effects on Bone Formation and Osteoblast Differentiation

The serum levels of osteocalcin—a marker of bone formation—were not affected by GCR in young adult mice (Figure 5A). Consistent with these changes, the mRNA levels of the osteoblastogenesis markers *Bglap* and *Akp* were unaffected in femur shafts from mice of GCR group (Figure 5B).

Bone marrow-derived stromal cells from irradiated mice cultured under osteogenic conditions exhibited no changes in osteoblast differentiation (Figure 5C). Taken together, these results demonstrate that exposure to GCR causes bone loss in young adult mice by increasing the function of osteoclasts, and Sirt3 may play a role in this effect.

### 2.4. GCR Increased Mitochondrial ROS but Did Not Affect Mitochondrial Mass

MitoSOX Red staining of the preOCs demonstrated a moderate but significant increase in steady-state levels of mitochondrial reactive oxygen species (ROS) in cells isolated from GCR exposed mice (Figure 6A,B). We have detected no changes when we measured mitochondrial mass using Mitotracker green staining in these cells (Figure 6A,B), suggesting exposure to GCR results in a persistent elevation in reactive species (presumably superoxide), which can be attributed to an altered mitochondrial function rather than an increased number of mitochondria since the mitochondrial mass in GCR or sham groups appeared not to differ.

### 2.5. GCR Increased Mitochondrial and Non-Mitochondrial Respiration

Because of its key role in OC differentiation and function, next, we determined the mitochondrial energy metabolism to assess whether GCR caused any changes in cellular and mitochondrial bioenergetics using Seahorse XF extracellular flux analysis (Figure 7) [64,67,68,69].

The increase in basal mitochondrial oxygen consumption rate (OCR) in the GCR group indicated an increased electron transport chain (ETC) activity in these cells. ATP-linked respiration was determined by treating cells with an ATP synthase (ETC Complex V) inhibitor, oligomycin. Increased levels of ATP-linked respiration suggest an increased energy demand in cells isolated from GCR exposed mice. At the same time, we noted a significant increase in OCR associated with proton leak implying GCR resulted in a disruption of electron flow through ETC, which was compensated by increased reserved respiratory capacity (difference between maximum OCR and basal OCR). This response was evident when the computation of coupling efficiency (ratio of ATP linked respiration/basal mitochondrial respiration) demonstrated no significant differences between sham-irradiated and GCR exposed groups. When ETC activity was halted by administering ETC Complex I and II blockers, Antimycin A and rotenone, the remaining non-mitochondrial oxygen consumption was also significantly higher in the GCR group, implying GCR has augmenting effects on cytosolic oxygen consuming enzymes.

### 2.6. GCR Induced Changes in Mitochondrial Metabolism in PreOCS

The determination of mitochondrial metabolic profile of preOCs of sham irradiated and GCR exposed mice in vitro, using a colorimetric kinetic assay produced intriguing data consistent with our results obtained from ROS measurements and extracellular flux analysis. We observed several significant differences in substrate utilization profiles between sham and GCR groups. Exposure to GCR significantly decreased the use of glycerol 3-phosphate, pyruvic acid, citric acid, isocitric acid, succinic acid, fumaric acid as well as palmitoyl carnitine and α-ketoisocaproic acid in permeabilized preOCs (Figure 8).

Glycerol 3-phosphate shuttle can direct cytosolic reducing equivalents to the mitochondrial oxidative phosphorylation pathway to generate ATP [70]. Furthermore, utilization of TCA cycle intermediates such as isocitric acid, and fumaric acid also generate NADH for driving the synthesis of ATP through oxidative phosphorylation [71]. Similarly, there was a significantly less utilization of α-ketoisocaproic acid, an intermediate metabolite of essential amino acid leucine, in the GCR exposed group’s preOCs. This intermediate regulates the ATP supply through acetyl CoA’s entry to the TCA cycle [71]. Thus, such decreases in the utilization of these substrates could indicate a probable allosteric downregulation of TCA cycle enzymes due to high levels of ATP production following GCR exposure in the preOCs [72]. Succinic acid not only provides FADH2 for ATP production via oxidative phosphorylation but also directly links the TCA cycle to mitochondrial ETC through succinate dehydrogenase complex (Complex II) [73,74]. Observed decreased utilization of succinic acid strongly suggests a GCR induced defect in Complex II, which could also be a plausible source for the persistent increase in mitochondrial ROS [73] that was measured in preOCs (Figure 6).

Interestingly, there was a statistically significant increase in glutamic acid consumption in preOC cells, which were generated from GCR exposed mice compared to that of the sham group (Figure 8). Glutamate has been shown to increase oxygen consumption in addition to altering mitochondrial fission–fusion dynamics [75] and hence, boosting ROS generation [75,76], which is consistent with our MitoSOX Red labeling (Figure 6) and Seahorse Extracellular Flux study results (Figure 7).

Ornithine consumption of preOCs was also increased in the GCR group (Figure 8). Because preOCs lack ornithine transcarbamylase, which converts ornithine to citrulline, ornithine was possibly utilized as a precursor for the biosynthesis of polyamines or proline and was not used to generate citrulline [77,78]. There were no significant changes in usage of glycolytic substrates in cells obtained from GCR exposed mice (Figure 8).

## 3. Discussion

Living in space is an amazing experience, but it would be taxing on astronauts’ health [79,80,81]. Long-duration space missions will expose astronauts to solar particle event (SPE) and galactic cosmic ray (GCR) irradiation, both of which are types of charged particle ionizing radiation [82]. Therefore, there are significant health concerns, including skeletal, cardiovascular, central nervous system injuries or increased cancer risk associated with long-term human space travel. A limited amount of research has been conducted to determine the mechanisms of bone loss associated with mixed field high LET space radiation. Thus, in this study, we examined the skeletal effects of simulated deep space GCR, using the recently developed simplified version of the NASA consensus formula of five different ions (^28^Si, ^4^He, ^16^O, ^56^Fe and ^1^H). Our data provide a snapshot of structural, molecular and metabolic changes that occur in bones of male adult mice 3 months after they were exposed to 0.5 Gy simplified GCR.

Bone is an incredibly dynamic tissue, with considerable continuous remodeling throughout its lifetime. For bone to function properly, bone formation by osteoblasts (OBs) and bone resorption by osteoclasts (OCs) must be tightly regulated [8,9]. Numerous in vitro, in vivo and human studies demonstrated both cell types contribute to low LET ionizing radiation induced bone loss in the context of clinical relevance [11,12,19,22,25,27,28,29,31,32,33,34,35,37,38,39,46]. However, because high LET space radiation has significantly distinct effects on normal tissues, extrapolating the results of low LET studies in terms of space travel induced bone loss has been somewhat difficult [22,40,41,42,43,44]. In addition, the use of different single high-energy particles and a wide range of doses also generated inconsistent results regarding the cell types and mechanisms contributing to the high LET radiation induced bone loss [22,40,41,42,43,44]. In the current study, we found that trabecular bone mass and the number was significantly decreased GCR exposed mice. This was consistent with findings from other groups [11,12,48]. Our data also demonstrated no changes in femoral cortical bone and no trabecular changes in the lumbar vertebrae L5 in these mice, which was contradictory to the findings from an earlier study [83]. Lang et al. (2004) reported a significant endocortical thinning in crewmembers after 4–6-month flights on the International Space Station, although similar to our results, they also found no significant compartment-specific bone loss in the spine. Interestingly, a very recent study exhibited significant decreases in trabecular bone volume fraction and thickness, with no changes in cortical porosity or thickness in the tibias of 17 astronauts following a >3-month mission [84]. However, they also reported a significant decrease in cortical volumetric bone density. The existence of another significant space hazard (i.e., microgravity) in both studies implies that the effects of GCR alone might differ and/or be exacerbated during deep space missions when it is combined with gravitational unloading.

The majority of the earlier ground-based space radiation studies conducted with single or mixed particles indicated that increased bone resorption by OCs significantly contributes to this unwanted health hazard [12,45,46,48,52]. OCs are multinucleated cells formed by the fusion of mononuclear progenitors that are differentiated from the monocyte/macrophage lineage of hematopoietic origin. This process is under the control of M-CSF (macrophage colony-stimulating factor) and RANKL (receptor activator of nuclear factor kappa-B ligand) [85]. After differentiating and attaching to the bone matrix, mature OCs undergo dramatic cytoskeletal reorganization and polarization to form the tight sealing zone and ruffled border for bone resorption of trabecular bone. This process produces distinct molecular signals that can be detected in serum. Our results demonstrated slightly increased (though not significant) serum levels of bone resorption marker CTx, which was also reported in astronauts as early as 15 days in space by Gabel et al. [84]. Furthermore, in agreement with previous findings of other groups [12,43,86] the expressions of late/terminal osteoclast differentiation markers, *Acp5* and *Ctsk* were significantly increased in bone shafts of our GCR exposed mice.

The dissolution of bone minerals and the digestion of organic bone matrix in the resorption lacuna are executed by hydrochloric acid and cathepsin K, which are released through lysosomes [87,88,89]. The transportation of lysosomes from the perinuclear region of OCs to the ruffled border is driven by cytoskeletal ATPases, relying heavily on the energy released from ATP hydrolysis for their activity [87,90]. This significant energy requirement for the bone-resorbing function of mature OCs is met by the electron transport chain and explains why OCs are rich in mitochondria [87,90]. Indeed, our extracellular flux analysis revealed an increase in basal and ATP-linked mitochondrial respiration in GCR group preOCs. The reserve respiratory capacity of the cells from the GCR group was also significantly elevated, suggesting an increase in the energy demanding bone resorption process in these cells. We have previously reported similar findings in an aging related bone loss study with Sirtuin 3 (Sirt3) knockout murine model [64]. Molecular or pharmacological inhibition of Sirt3 resulted in preventing bone loss via decreased OC’s respiratory activity and osteoclastogenesis [64]. The current study also revealed that mRNA expression of *Sirt3* was significantly increased in the GCR exposed OCs. As the major mitochondrial deacetylase, Sirt3 is a stress-response protein, which augments intracellular antioxidant activity to combat ionizing radiation induced ROS [91]. As our data exhibited, mitochondrial steady-state ROS levels were increased in preOCs differentiated from the GCR group’s bone marrow macrophages at a 3-month time point, strongly suggesting that Sirt3 might be responding to GCR induced persistent oxidative stress but in turn over activating the formation and activity of OCs. Moreover, an additional push for osteoclastogenesis via higher ROS levels appears to exist in the irradiated mice, which was previously validated as ROS produced by osteoclasts stimulate and facilitate resorption of bone tissue [11,56].

Increased mitochondrial respiration, as well as mitochondrial ROS levels, led us to investigate whether GCR further amplified the mitochondrial biogenesis. We measured mRNA levels of transcription of *Pgc1β* and found no changes compared to that of the sham group. This result was also confirmed when we evaluated the mitochondrial mass in preOCs via Mitotracker Green labeling of the cells. Taken together, these results insinuated that GCR induced alterations in redox homeostasis of OCs contribute to the upregulation of their mitochondrial activity, thus their bone resorption function.

After unveiling such changes in mitochondrial respiration and redox homeostasis of OCs in the GCR group in our study, the next logical step was to explore if these alterations were accompanied by any changes in the fuel metabolism of these cells. Mitochondria are capable of transitioning between alternative substrates to fuel metabolism according to cells’ needs [92]. Recent reports also pointed out mitochondria as a major hub of global metabolic changes associated with space travel [93]. Our results indicated several significant differences in utilization of metabolic substrates between sham and GCR groups’ preOCs. Because ATP is required for increased osteoclastic bone resorption activity, we initially anticipated that use of the substrates that would feed into ATP generating pathways would be amplified in GCR exposed group. Interestingly we found that glycerol 3-phosphate, α-ketoisocaproic acid, pyruvic acid, isocitric acid, fumaric acid, succinic acid were used less by preOCs from the GCR group compared to sham preOCs. All these substrates can help ATP production by generating the reducing equivalents to run oxidative phosphorylation via either TCA cycle activity or directly shuttling them to the electron transport chain [70,71]. Although it initially appears counterintuitive, our results could be explained by the necessity of a healthy crosstalk between redox and energy metabolisms for cellular function. Our results showing increased mitochondrial ROS levels persistent 3 months following GCR exposure, coupled with the decreased use of succinic acid, could indicate a defect in ETC Complex II (succinate dehydrogenase complex). As we previously reported, Complex II could be a significant source of mitochondrial ROS [73]. It is also possible that the cells were attempting to downregulate a non-physiological surge in their ATP production, indicated by increased ATP-linked respiration, by reducing their TCA cycle activity. It has been shown that when in excess, ATP levels could hinder cell growth due to limited divalent cation levels. For example, Mg^2+^ can act as a counter-ion for ATP during protein synthesis [94].

In contrast to decreased utilization of multiple TCA intermediates, ornithine and glutamate utilization by preOCs were both increased in the GCR group compared to sham. Increased glutamate use could be a double edge sword in tissues. Because of its critical role in the metabolism of mammalian cells, glutamate could be used for nitrogen assimilation, amino acid biosynthesis and incorporated into redox metabolism by being a precursor for glutathione [75,76]. Because there was no change in glutamine utilization between the GCR and sham groups, we think the conversion of glutamate to glutamine is not the major reason for its increased use. In the context of our other findings in this study, we believe the increased glutamate utilization could in part be responsible for increased levels of ROS and oxygen consumption that were observed in preOCs from the GCR group. Such effects of glutamate have been previously reported in different cell and tissue types [75,76]. Slightly but significantly increased ornithine use was also somewhat unexpected. Because preOCs do not possess the enzyme necessary for the carbamylation of ornithine, its conversion to citrulline was less likely. The possible pathways for ornithine’s fate could be its use in the biosynthesis of polyamines and/or proline [77,78] but this speculation still needs to be investigated further in preOCs.

In conclusion, the current study presents evidence for possible molecular and metabolic mechanisms by which high energy particle radiation—in the form of simulated GCR—causes trabecular bone loss. While our study design has limitations, the results we presented in this paper demonstrate the importance of metabolic rewiring of OC mitochondria in skeletal health during deep space missions. We recognize that (1) single dose delivered at one time instead of chronic or fractionated fashion, (2) single time point following exposure for tissue collection instead of a longitudinal assessment of multiple time points, (3) absence of a gravitational unloading group (alone or combined with GCR exposure), and (4) use of only male mice without addressing any possible sex differences must be taken into consideration while interpreting our data. However, parallel findings recently reported by others also suggest that complex and dynamic redox state and metabolic reprogramming of OC may be evaluated as viable targets when developing countermeasures against adverse effects of space travel.

## 4. Materials and Methods

### 4.1. Animals and Simulated GCR Exposure

Eight weeks old male Balb/C mice were purchased from The Jackson Laboratory (Bar Harbor, ME, USA) and housed five animals per cage. The mice received standard rodent chow and water ad libitum until they reached 6 months of age. Once the animals were 6 months old, they were transferred to Brookhaven National Laboratory (BNL), where they were acclimated for 1 week before they were exposed to a simplified GCR simulation beam designed by NASA. The beam consists of protons at 1000 MeV, ^28^ Si at 600 MeV/n, ^4^He at 250 MeV/n, ^16^O at 350 MeV/n, ^56^Fe at 600 MeV/n and protons at 250 MeV. The total dose of 0.5 Gy was chosen as a probable equivalent dose that an astronaut would receive during a 1.5–2-year deep space mission [95]. Sham irradiated mice received the same protocol but did not receive any radiation. Two days after total body simulated GCR exposure both irradiated and sham treated mice were shipped back to UAMS. During 8-week quarantine protocol mice received 150-ppm fenbendazole in chow. At the end of the quarantine period, mice were transferred to non-barrier animal facility and maintained an additional 4 weeks until they were euthanized for blood and tissue collection.

### 4.2. Micro-CT Analysis

Bone architecture was determined on dissected femora and lumbar vertebra (L5) cleaned of adherent tissue, fixed in Millonig’s phosphate buffer (Leica Microsystems, Buffalo Grove, IL, USA) and stored in 100% ethanol. Bones were scanned with μCT40 (Scanco Medical, Bruttisellen, Switzerland) at high resolution for obtaining images and at medium resolution for making quantitative determinations as described previously [96]. Scans were performed at medium resolution (12 µm isotropic voxel size) for quantitative determinations. For the latter, a Gaussian filter (sigma = 0.8, support = 1) was applied. Scanco Eval Program v.6.0 was used for measuring bone volume. Scan settings were applied to X-ray tube potential (E = 55 kVp), X-ray intensity (I = 145 μA) and integration time (220 ms). Nomenclature conforms to recommendations of the American Society for Bone and Mineral Research [97]. Cortical dimensions were determined at both the diaphysis (18 slices, midpoint of the bone length as determined in scout view) and the metaphysis (starting 8–10 slices away from the growth plate so as to avoid the growth plate and proceeding proximally for 50 slices) at a threshold of 200 mg/cm^3^. Trabecular bone measurements of the distal femur were made on 151 transverse slices that were taken from the epicondyles, extending toward the proximal end of the femur; cortical bone and primary spongiosa were manually excluded from the analysis. Trabecular architecture was determined using sphere filling distance–transformation indices without assumptions about the bone shape as a rod or plate. The fifth lumbar vertebra (L5) was scanned from the rostral growth plate to the caudal growth plate to obtain 233 slices. BV/TV in the vertebra was determined using 100 slices (1.2 mm) of the anterior (ventral) vertebral body immediately inferior (caudal) to the superior (cranial) growth plate. All trabecular analyses were performed on contours of every 10 to 20 cross-sectional images and were measured at a threshold of 220 mg/cm^3^.

### 4.3. Bone Histology

The terminology used in this report is that which is recommended by the Histomorphometry Nomenclature Committee of the American Society for Bone and Mineral Research [97]. Femurs were fixed for 24 h in Millonig’s 10% formalin followed by dehydration and embedded undecalcified in methyl methacrylate. Five-μm-thick longitudinal sections were cut in the medial-lateral plane. The sections were stained with naphthol AS-MX and Fast Red TR salt (Sigma-Aldrich, St. Louis, MO, USA) to detect osteoclast formation. Sections also were stained with 0.3% toluidine blue in phosphate buffered citrate, pH 3.7, to visualize osteoblasts, osteoid and cement lines. Standard histomorphometric parameters were measured using an Olympus BX53 microscope and Olympus DP73 camera (Olympus America, Inc., Waltham, MA, USA) interfaced with a digitizer tablet with OsteomeasureTM software version 4.1.0.2 (OsteoMetrics Inc., Decatur, GA, USA). One section per sample was analyzed by a histopathologist blinded to the study groups.

### 4.4. CTx and Osteocalcin ELISA

Blood was collected into 1.7 mL microcentrifuge tubes by retro-orbital bleeding. Blood was then kept on ice for 30 min and centrifuged at 10,000× *g* rpm for 10 min to separate serum from cells before analysis. Circulating CTx and osteocalcin in serum was measured using a mouse RatLaps (CTx-I) EIA kit (Immunodiagnostic Systems, Boldon, United Kingdom) and Osteocalcin enzyme immunoassay kit (Thermo Fisher, Waltham, MA, USA) according to the manufacturer’s directions.

### 4.5. Osteoclast Differentiation

Bone marrow macrophages were obtained as described previously [64]. Briefly, whole bone marrow cells were cultured with 10% FBS, 1% PSG overnight in the presence of 10 ng/mL of M-CSF (R&D Systems, Minneapolis, MN, USA). Non-adherent bone marrow cells were collected and cultured in Petri dishes with 30 ng/mL of M-CSF and adherent bone marrow macrophages were harvested as osteoclast progenitors 4 days later. To generate mature osteoclasts, bone marrow macrophages were cultured with 30 ng/mL of M-CSF and 30 ng/mL of RANKL (R&D Systems) for 5 days. To enumerate osteoclasts, the cells were stained for tartrate-resistant acid phosphatase (TRAP), using the Leukocyte Acid Phosphatase Assay Kit, following the manufacturer’s instructions (Sigma-Aldrich). An osteoclast was identified as multinucleated (>3 nuclei) TRAP-positive cells.

### 4.6. Osteoblast Differentiation

Bone marrow macrophages were obtained as described previously [64]. Briefly, whole bone marrow cells were cultured with 50 μg/mL of ascorbic acid (Sigma-Aldrich) in 100 mm dishes for 6 days. Adherent bone marrow stromal cells were then re-plated in triplicate in 12 well plates with 50 μg/mL of ascorbic acid and 10 mM beta-glycerophosphate (Sigma-Aldrich). The mineralized matrix was stained with 40 mM Alizarin Red solution, following the manufacturer’s instructions (Sigma-Aldrich).

### 4.7. Quantitative RT-PCR

Total RNA was purified from cultured osteoclasts or bone tissues using TRIzol reagent (ThermoFisher Scientific) according to the manufacturer’s directions. RNA was quantified using a Nanodrop instrument (ThermoFisher Scientific) and 1–2 µg of RNA was then used to synthesize cDNA using a High-Capacity cDNA Reverse Transcription kit (Applied Biosystems) according to the manufacturer’s instructions. Transcript abundance in the cDNA was measured by quantitative PCR using TaqMan Universal PCR Master Mix (ThermoFisher Scientific). The primers and probes for murine *Acp5* (Mm00432448_m1), *Ctsk* (Mm00484039_m1), *Bglap* (Mm03413826_mH), *Akp* (Mm00475831_m1), *Sirt3* (Mm00452131_m1), *Nfatc1* (Mm00479445_m1) and *Pgc1β* (Mm00504720_m1) were manufactured by the TaqMan^®^ Gene Expression Assays service (Applied Biosystems). Relative mRNA expression levels were normalized to the house-keeping gene ribosomal protein S2 (Mm00475528_m1) using the ΔCt method.

### 4.8. Mitochondrial ROS and Mitochondrial Mass

Bone marrow macrophages were plated in 4-well tissue culture chamber slides and treated with 30 ng/mL RANKL for 3 days to obtain preOCs. The media in the wells was replaced with 5 mM pyruvate containing DPBS once and then labeled with Mitotracker Green (100 nM, in 0.1% DMSO; ThermoFisher Scientific) or MitoSOX Red (2 μM, in 0.1% DMSO; ThermoFisher Scientific) for 15 min at 37 °C. For positive control cells were treated with 10 μM Antimycin A during labeling. After labeling, cells were washed with DBPS and kept on ice. Sample images were acquired using EVOS fluorescence microscope (531/40 nm Excitation; 593/40 nm Emission and 482/25 nm Excitation; 524/24 nm Emission filters) and analyzed using ImageJ software [69,98].

### 4.9. Mitochondrial Respiration and Cellular Bioenergetics

Bone marrow macrophages were plated in Seahorse XF96 plates and treated with 30 ng/mL RANKL for 3 days to obtain preOCs. The media in the wells was replaced with XF assay media (Agilent), and the plate was kept in a non-CO_2_ incubator for 20 min at 37 °C. Cellular respiration measurements were recorded with the Seahorse XF96 analyzer (Agilent), before 10 μg/mL oligomycin (Sigma-Aldrich) was added to inhibit mitochondrial ATP synthase to measure the decrease in the OCR that is linked to mitochondrial ATP generation. An oxidative phosphorylation uncoupler, FCCP (10 μM, Sigma-Aldrich) was used to assess the maximal respiration potential of the cells. The amount of nonmitochondrial oxygen consumption was determined by inhibiting the electron respiratory chain activity using antimycin A and rotenone cocktail (10 μM, Sigma-Aldrich). These data were used to calculate the mitochondrial basal respiration, ATP-linked respiration, reserve respiratory capacity, proton leak and coupling efficiency as we previously described [61,67,69].

### 4.10. Mitochondrial Function and Substrate Utilization

Bone marrow macrophages were obtained as described previously [64]. The cells were plated in 96-well plates and treated with 30 ng/mL RANKL for 3 days to obtain preOCs. Mitochondrial metabolic S-plates (Biolog, Hayward, CA, USA), which contain different substrates, were used for the screening. Briefly, assay mix containing a mitochondrial assay buffer solution and a redox MC dye (Biolog) were pipetted into all substrate containing wells of the S-plates and incubated at 37 °C for 1 h to allow substrates to fully dissolve. 100 µg/mL saponin was added before the substrate solutions were transferred into the 96-well plates containing the preOCs. The color formation was read kinetically on a microplate reader using OD_590_ for 4 h. The rates were plotted and calculated for the linear range [99].

### 4.11. Statistical Analysis

All data were analyzed using GraphPad Prism 9 (GraphPad Software). For all graphs, data are represented as the mean ± SD unless otherwise specified. For comparison of two groups, data were analyzed using a two-tailed Student’s *t*-test, and for comparison of 3 groups, one-way ANOVA with Tukey’s post-hoc test was used. *p* < 0.05 was considered significant.

## Figures and Tables

**Figure 1 ijms-22-11711-f001:**
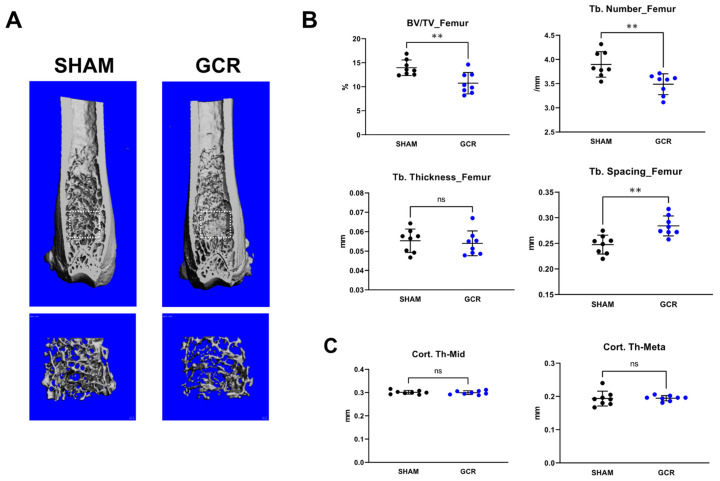
GCR induced trabecular bone loss in adult male mice 3 months after exposure. (**A**–**C**) Imaging and quantification of femoral bones from sham and GCR exposed male mice by micro-CT after sacrifice (n = eight animals/group). (**A**) Representative images of the femoral bone. (**B**) Trabecular bone measurements of the distal femur were made on 151 transverse slices that were taken from the epicondyles, extending toward the proximal end of the femur; cortical bone and primary spongiosa were manually excluded from the analysis. Trabecular bone parameters demonstrated significant decreases in the percentage of trabecular bone volume to tissue volume (BV/TV) and trabecular number (Tb. Number) and a significant increase in trabecular spacing (Tb. Spacing). (**C**) No significant changes were observed in cortical thickness in midshaft (Cort. Th-Mid) or metaphysis (Cort. Th-Meta). Data are presented as mean ± SD. *p* values were determined using Student’s *t*-test. (n = 8, ** *p* < 0.01., ns: not significant).

**Figure 2 ijms-22-11711-f002:**
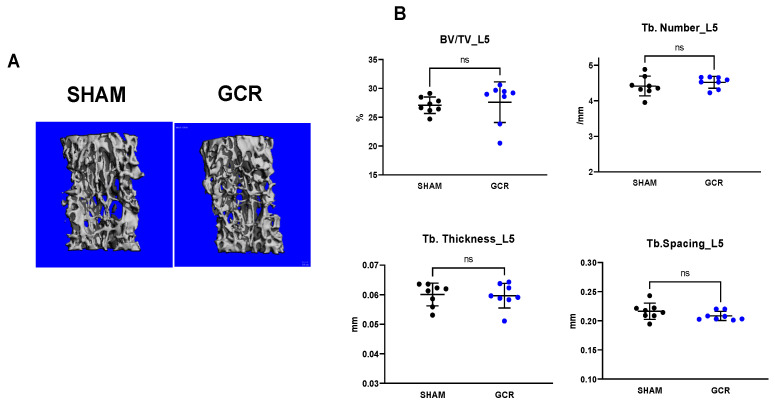
GCR did not alter spinal trabecular bone in male mice. (**A**) Representative images of lumbar vertebra (L5) of sham and 0.5 Gy GCR exposed adult male BalbC mice 3 months following exposure. (**B**) No significant changes were detected in trabecular parameters (BV/TV; Tb. number, Tb. spacing, Tb. Thickness) in L5 vertebra (n = eight mice per group). Data are presented as ± SD. *p* values were determined using Student’s *t*-test. (n = 8, ns: not significant).

**Figure 3 ijms-22-11711-f003:**
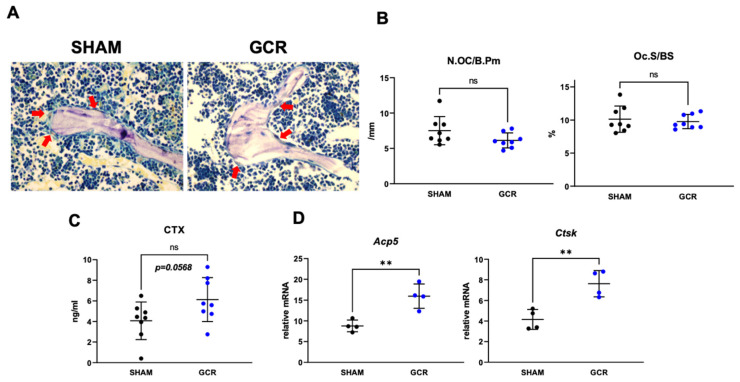
Late OC differentiation markers in serum were increased when mice were exposed to 0.5 Gy GCR. (**A**,**B**) GCR or sham exposed adult male BalbC mice did not show any differences in the number (N.OC/BPm; osteoclast number per bone perimeter) of OCs on the endocortical bone surface 3 months after GCR exposure. Femoral trabecular bone histology measurements also revealed no significant changes in the surface area (Oc.S/BS; osteoclast surface per bone surface) of OCs between sham and GCR exposed mice. (**C**) Bone resorption marker CTx levels were measured in serum and found to be slightly higher in GCR exposed mice. (**D**) Late OC differentiation markers cathepsin K (*Ctsk*) and tartrate-resistant acid phosphatase type 5 (*Acp5*) were determined in the whole femoral bones and were found to be significantly elevated in mice at 3 months following 0.5 Gy GCR exposure. Data are presented as ± SD. *p* values were determined using Student’s *t*-test. (n = 4, ** *p* < 0.01, ns: not significant).

**Figure 4 ijms-22-11711-f004:**
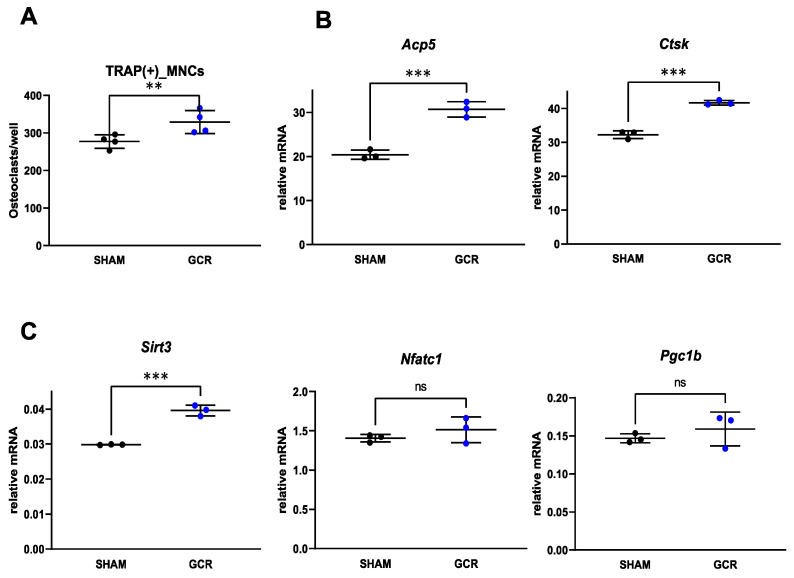
OC differentiation was increased when bone marrow macrophages were isolated from sham or GCR exposed mice in vitro. (**A**) Bone marrow macrophages were isolated from 9-month-old male BalbC sham or GCR exposed mice 3 months after the mice were irradiated and cultured with M-CSF (30 ng/mL) and RANKL (30 ng/mL) for 5 days. The number of TRAP+ multinucleated osteoclasts was significantly higher GCR group. (**B**) Expression of OC differentiation markers cathepsin K (*Ctsk*) and tartrate-resistant acid phosphatase type 5 (*Acp5*) were found to be significantly increased in the GCR group. (**C**) No significant differences were noted between sham and GCR groups when mRNA levels of early OC differentiation marker *Naftc1* or in mitochondrial biogenesis transcription factor *Pgc1β* were measured in vitro. However, expression of major mitochondrial deacetylase *Sirt3*, which plays an important role in OC function, was significantly increased in the GCR group in vitro. Data are presented as ± SD. *p* values were determined using Student’s *t*-test. (n = 3, ** *p* < 0.01, *** *p* < 0.001. ns: not significant).

**Figure 5 ijms-22-11711-f005:**
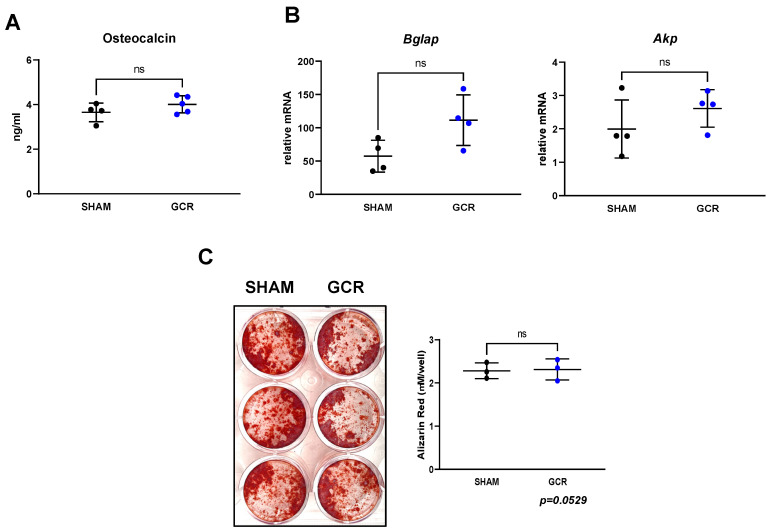
Osteoblast differentiation was not affected by GCR exposure at 3 months. (**A**) Serum marker of OB differentiation, osteocalcin, was measured by ELISA and found to be unchanged in GCR exposed mice (compared to the sham-irradiated group). (**B**,**C**) Bone marrow stromal cells were isolated from 9-month-old mice 3 months after 0.5 Gy GCR exposure and cultured with ascorbate (50 mg/mL) and β-glycerophosphate (10 mM) for 14 days for OB differentiation. Assessment of neither mRNA levels of OB differentiation markers (**B**) nor the number of differentiated OBs in vitro (**C**) demonstrated any significant differences between sham-irradiated and GCR exposed groups. Representative images of Alizarin Red stained OBs can be seen in panel C. Data are presented as ± SD. *p* values were determined using Student’s *t*-test. (n = 3–4, ns: not significant).

**Figure 6 ijms-22-11711-f006:**
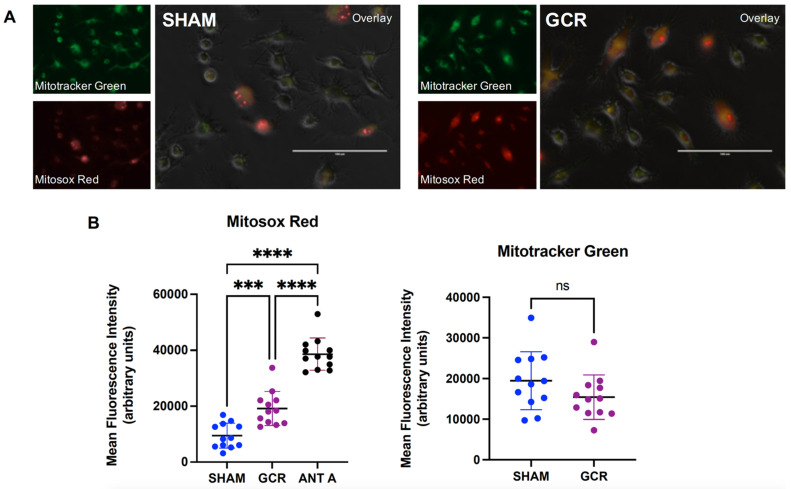
Exposure to 0.5 Gy GCR increases mitochondrial ROS in vitro. Bone marrow macrophages were isolated from 9-month-old male BalbC sham or GCR exposed mice 3 months after the mice were irradiated and cultured with M-CSF (30 ng/mL) and RANKL (30 ng/mL) for 3 days. Cells were simultaneously stained with 2 μM MitoSOX Red (upper left images) and 100 nM MitoTracker Green (bottom left images) for 20 min. (**A**) The epifluorescent pictures of randomly chosen 12 areas were then taken using a fluorescent microscope in a blinded fashion. Antimycin A (10 μM) was used as the positive control. (**B**) The images were also analyzed with ImageJ software in a blinded fashion. Each area had between 18–53 cells and the fluorescent intensities of each cell were measured. MitoTracker Green and MitoSOX results were presented as mean fluorescent intensity (MFI) per cell in arbitrary units. Data are presented as ± SEM. *p* values were determined using Student’s *t*-test between the two groups (n = 12, ns: not significant). When there were 3 groups, one-way ANOVA with Tukey’s post-hoc test was utilized (n = 12, *** *p* < 0.001, **** *p* < 0.0001).

**Figure 7 ijms-22-11711-f007:**
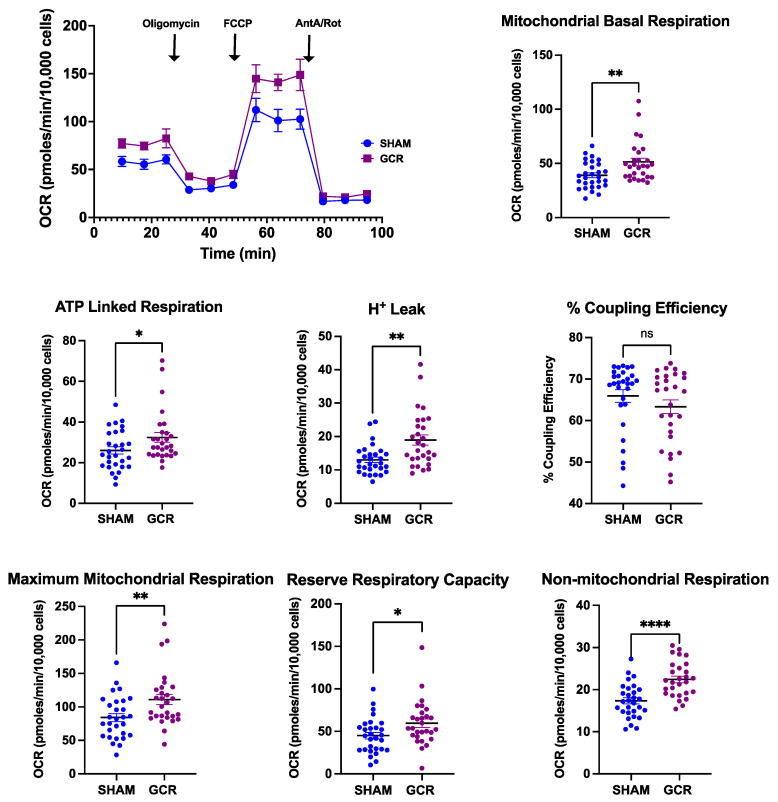
GCR exposure increases overall mitochondrial and non-mitochondrial respiration. Bone marrow macrophages were isolated from 9-month-old male BalbC sham or GCR exposed mice 3 months after the mice were irradiated and cultured with M-CSF (30 ng/mL) and RANKL (30 ng/mL) for 3 days. Seahorse extracellular Flux analysis revealed moderate but significant increases in basal mitochondrial and non-mitochondrial respiration in pOCs isolated from GCR exposed mice. ATP-linked respiration and maximum mitochondrial oxygen consumption were also significantly increased in the GCR group compared to the sham irradiated group. Although there was an increase in proton leak in pOCs isolated from GCR exposed mice, no differences were detected in coupling efficiency between the groups. Data are presented as ± SD. *p* values were determined using Student’s *t*-test. (n = 28–30 wells, * *p* < 0.05, ** *p* < 0.01, **** *p* < 0.0001, ns: not significant).

**Figure 8 ijms-22-11711-f008:**
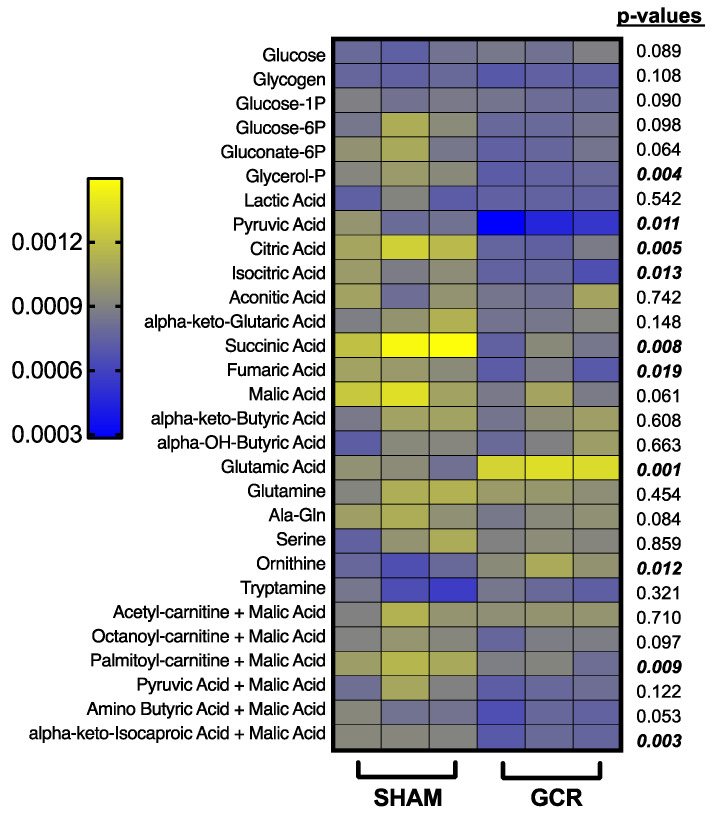
GCR altered the substrate utilization of preOCs in vitro suggesting exposure to high LET IR can result in mitochondrial metabolic reprogramming. Bone marrow macrophages were isolated from 9-month-old male BalbC sham or GCR exposed mice 3 months after the mice were irradiated and cultured in 96-well plates with M-CSF (30 ng/mL) and RANKL (30 ng/mL) for 3 days. Substrate utilization of cells was kinetically determined by transferring multiple NADH and FADH2 producing substrates from Biolog metabolic S-plates. The linear range of the OD590 measurements was used to calculate the changes in color formation in the presence of a redox MC dye using a standard plate reader. (n = 3 per substrate per experiment and the experiment was repeated twice with similar results). Data are presented as ± SD. *p* values were determined using Student’s *t*-test between the two groups and listed in the figure for each substrate.

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
