# Peer review of "Simulated Galactic Cosmic Rays Modify Mitochondrial Metabolism in Osteoclasts, Increase Osteoclastogenesis and Cause Trabecular Bone Loss in Mice"

_ijms, 2021, doi:10.3390/ijms222111711_

Round 1

Reviewer 1 Report

Summary:

In this manuscript, the authors tested the effect of low dose high linear energy transfer radiation with a combined ions of (28Si, 4He, 16O, 56Fe and 1H) on 6-month-old male mice to examine the galactic cosmic rays (GCR) radiation induced bone loss on astronauts with a focus on osteoclasts. Using serum, tissue and cells harvested 3 months after the radiation, the authors found reduction in femoral trabeculae, while the cortical bone thickness and the vertebral trabeculae were not affected. Inconsistent results were reported, with increased osteoclast markers in cortical bone but no difference of the N.OC/B.Pm and Oc.S/BS were found within the trabecular region, in contradiction with the microCT results. Further interesting analysis of OC mitochondrial function with fluorescent markers and cell substrate analysis demonstrated altered mitochondria metabolic pathways. No difference is found in osteoblast markers, without confirmation of the dynamic histomorphometry analysis.

Comments:

Overall, this manuscript offers nice and detailed mitochondria analysis in OCs but is missing supporting stained tissue sections with a clear description of the region of interest. Discussion is needed for the contradicting results. It needs to be revised concerning the following questions.

  1. Method description of GCR radiation on mice was not clear. Was the radiation applied to the whole body or localized regions? Is there a radiation difference between regions of femur and vertebra? Figure 1, A shows a more localized reduction of BV/TV in the ROI, indicated by the white box. However, for the more proximal regions of the GCR group femur (above the white box), the BV/TV reduction is not as apparent between the GCR and SHAM group.
  2. Description about the region of interest in tissue sections is not clear. Were N.OC/B.Pm and Oc.S/BS quantified at two different regions or not (one on the endocortical bone surface and trabecular surface)? Besides, in microCT results, lower BV/TV in trabecular bone and no difference in cortical thickness were found in GCR group femur. But later examination showed that late/mature osteoclast markers are elevated in cortical bone and cell culture, and the N.OC/B.Pm and Oc.S/BS in the trabecular region are not different. No explanation is provided regarding the inconsistence of OC or OC markers and the microCT results.

Specific comments:

  1. Italicize “in vitro” and “in vivo” in the context, e.g., page 2, line 45
  2. Page 3, figure 1, caption missing word “mean”. Data are presented as mean+
  3. Page 4, figure 3, please provide representative images of the stained tissue sections alongside the scatter plot.
  4. Page 6, figure 5, make (A) bold to keep consistent.
  5. Page 7, figure 6, caption not properly including and explaining panel A and B. Significance marker missing from Panel B
  6. Page 11, line 342, refs are missing as indicated in the brackets.
  7. Page 13, line 463, ref not correctly inserted.
  8. Dynamic histomorphometry analysis should be considered when evaluating OB function.

Author Response

Authors would like to thank you for your detailed corrections and suggestions.  Please see the attachment for our responses to your review.

Reviewer 2 Report

The authors proposed an interesting study underlying the effects of simplified GCR exposure on skeletal health in vivo. the paper is well written and easy to follow. 

Figure 4C: the authors showed that the treatment does not affect the levels of Nfatc1. however this is an early osteoclastogenic markers that is induced in the early phase of osteoclast differentiation. It is not clear to me the meaning of assessing its levels since the have already demonstrated the modulation of other genes. 

the authors clearly reported the major limitations of this study. In fact, the effects on female mice should be addressed. 

Did the authors analyze the effects of anti resorptive drugs in their experimental conditions? 

Author Response

Authors would like to thank you for your suggestions.  Please see the attachment for our responses to your review.
